# Design and Characterization of a Novel Tool for the Antigenic Enrichment of *Actinobacillus pleuropneumoniae* Outer Membrane

**DOI:** 10.3390/pathogens9121014

**Published:** 2020-12-02

**Authors:** Fabio Antenucci, Armen Ovsepian, Agnieszka Wrobel, Hanne Cecilie Winther-Larsen, Anders Miki Bojesen

**Affiliations:** 1Department of Veterinary and Animal Sciences, University of Copenhagen, Stigbøjlen 4, 1870 Frederiksberg C, Copenhagen, Denmark; fante@sund.ku.dk (F.A.); armovse@hotmail.com (A.O.); 2Section of Pharmaceutical Biosciences, Centre of Integrative Microbial Evolution, Department of Pharmacy, University of Oslo, Sem Sælandsvei 3, 0316 Oslo, Norway; agnieszka_wrobel1@wp.pl (A.W.); h.c.winther-larsen@farmasi.uio.no (H.C.W.-L.)

**Keywords:** *Actinobacillus pleuropneumoniae*, vaccine development, immunogen screening, antigenic enrichment

## Abstract

Production and isolation of recombinant proteins are costly and work-intensive processes, especially in immunology when tens or hundreds of potential immunogens need to be purified for testing. Here we propose an alternative method for fast screening of immunogen candidates, based on genetic engineering of recombinant bacterial strains able to express and expose selected antigens on their outer membrane. In *Actinobacillus pleuropneumoniae*, a Gram-negative porcine pathogen responsible for extensive economic losses worldwide, we identified a conserved general secretion pathway (GSP) domain in the N-terminal part of the outer membrane protein ApfA (ApfA stem: ApfA_s_). ApfA_s_ was used as an outer membrane anchor, to which potential immunogens can be attached. To enable confirmation of correct positioning, ApfA_s,_ was cloned in combination with the modified acyl carrier protein (ACP) fluorescent tag ACP mini (ACP_m_) and the putative immunogen VacJ. The chimeric construct was inserted in the pMK-express vector, subsequently transformed into *A. pleuropneumoniae* for expression. Flow cytometry, fluorescence imaging and mass spectrometry analysis were employed to demonstrate that the outer membrane of the transformed strain was enriched with the chimeric ApfA_s_-ACP_m_-VacJ antigen. Our results confirmed correct positioning of the chimeric ApfA_s_-ACP_m_-VacJ antigen and supported this system’s potential as platform technology enabling antigenic enrichment of the outer membrane of *A. pleuropneumoniae*.

## 1. Introduction

Since its inception, genetic engineering has enabled protein expression and purification in a variety of microorganisms [1,2]. One of the possible applications of this technique is the expression of recombinant antigens for vaccine development, defined as immunogens. In the case of bacterial vaccines, these antigens are often proteins located on the outermost surface of the bacteria [3,4].

The identification of useful protein immunogens is not an easy task, despite the contribution offered by the advent of bioinformatic screening tools for these purposes [5,6,7]. One of the main challenges is the identification of conserved proteins able to offer protection against ideally all known variants of the bacterial species/types targeted [8,9]. Even when conserved candidates are successfully identified, prediction tools offer disappointingly little help regarding the actual potential of these proteins for vaccine development, often leading to a high degree of negative results when tested in vivo [10,11]. This in turn increases the number of candidates that need to be expressed and tested and hence the need for expression systems requiring less resources.

One option to account for this need is represented by antigenic enrichment, where the presence of a desired immunogen in a specific subcellular compartment, such as the bacterial outer membrane (OM), is increased [12,13]. Using that approach it is possible to engineer bacterial clones by heterologous gene transfer (HGT) to express recombinant proteins on the OM. The clones can then be administered in vivo to verify the protective potential of the candidate immunogens. This eliminates the need of isolating and purifying each of the proteins to be tested, thus reducing considerably the costs and workload needed [14]. To achieve targeted enrichment, the subcellular localization of the recombinant protein expressed often needs to be specifically engineered by introducing a short sequence of amino acids at the proteins N-terminal, defined as signal peptide [15]. It must be noted though that the presence of a correct signal peptide is often not sufficient to guarantee the correct subcellular localization of a recombinant protein and additional parameters need to be considered during the engineering process. This is particularly true in the case of secreted or membrane-bound immunogens, where incorrect folding, surface charge and cleavage can prevent access to the channels responsible for trans-membrane translocation, hindering the whole process as a result [16].

Once engineered, the recombinant protein is then expressed in the host organism of choice, where its correct subcellular localization needs to be confirmed. This is usually accomplished by using different reporter systems. One such system involves the utilization of monoclonal antibodies for indirect immunodetection [17]. The main disadvantage of this system is that in many instances specific antibodies binding the expressed protein are not available. This is especially relevant for proteins that have not yet been fully described, as is often the case for putative vaccine candidates. Alternatively, recombinant proteins can be labelled by using a wide array of fusion tags [18], facilitating not only detection but also increasing solubility and levels of expression [14]. Autofluorescent or immunogenic tags can be included in the sequence of the recombinant protein to allow detection and quantification but these tags tend to be rather large, thus interfering with the protein exportation through bacterial membranes and may significantly alter its antigenicity [14,19]. Both these problems can be addressed by using smaller tags for indirect fluorescent detection, namely tags recognized by specific enzymes able to mediate the post-translational labelling of the recombinant protein with a fluorescent substrate [20].

An additional factor to consider is how to ensure the stable integration of the recombinant protein into the OM. Most antigens of interest possess lipophilic domains and are naturally embedded into the outer layer of the OM, whereas others are secreted and require modifications to ensure OM localization. OM domains can be identified in several bacterial species by analyzing the mechanisms of transport and sequence of well-characterized OM determinants, such as fimbriae or lipoproteins [7,21,22]. Once identified, these domains can be included into the open reading frame (ORF) of a recombinant protein by polymerase chain reaction (PCR)-based tagging [23], enhancing the chances of its stable integration on the outer layer of the OM when exported [24]. 

Resuming, there are four crucial features that need to be included in a recombinant immunogen targeted for native OM localization: (i) a signal peptide for OM localization; (ii) an OM domain to anchor the protein to the bacterial surface; (iii) a detection domain for localization and quantification; (iv) a domain corresponding to the actual immunogen to be tested, either as a complete or partial protein. 

Here we describe the development of a novel platform technology according to these features that allows selective antigen-enrichment of the OM of *A. pleuropneumoniae*, a Gram-negative bacterium of great veterinary interest for which increasingly effective vaccines are needed [25,26,27]. Main aim of this study was to provide an initial assessment of the feasibility of antigenic enrichment of *A. pleuropneumoniae* OM in a quantifiable and cost-effective manner.

## 2. Results

### 2.1. ApfA_s_ is a Conserved OM Domain

Analysis by PANTHER™ identified amino acid residues 14–67 of the ApfA protein (ApfA_s_) as a conserved domain of general secretion pathway (GSP) proteins (PANTHER ID: PTHR30093). Further in silico analysis corroborated the inferred OM localization of ApfA (PSORTb 3.0), also supported by the prediction of a cytoplasmic domain at the N-terminal of ApfA (residues 1-11; PHOBIUS) and the folding of residues 16-66 to form a hydrophilic alpha-helix (I-TASSER). 

Analysis by protein BLAST (non-redundant protein sequences database) revealed a significant homology of the ApfA_s_ domain with peptidase-dependent pilins from several *Enterobacteriaceae* species. The following entry is reported here as representative of the analysis: MULTISPECIES: prepilin peptidase-dependent pilin [Enterobacteriaceae] (NCBI accession number: WP_000360918.1; 82% query coverage, 51,79% identity).

No relevant homology was detected between the ApfA_s_ domain and the following *E. coli* proteins—OmpA (GenBank accession number: NP_415477.1); LamB (GenBank accession number: VWQ00259.1); PhoE (GenBank accession number: VWQ01102.1); FimH (GenBank accession number: NP_418740.1); PapA (GenBank accession number: AAL67417.1); Flagellin (GenBank accession number: WP_001556318.1); AidA (GenBank accession number: ADD91708.1).

### 2.2. ApfA_s_-ACP_m_-VacJ is Expressed and Localized on A. pleuropneumoniae OM

Fluorescence imaging indicated that pMK_*apfa*_s_-ACP_m_-*vacJ* cells emitted higher fluorescent signal as compared to the wild type (wt) (Figure 1) when AcpS and fluorescent coenzyme A (CoA) are provided in solution. Under similar conditions, flow cytometry analysis detected a significant increase (*p* < 0.01) between 30–60% of the geometric mean of the emitted green fluorescence signal from the pMK_*apfa*_s_-ACP_m_-*vacJ* cells when compared to the wt (Figure 2). 

Sodium dodecyl sulfate polyacrylamide gel electrophoresis (SDS-PAGE) analysis showed no qualitatively detectable differences in banding patterns of the OM fractions of pMK_*apfa*_s_-ACP_m_-*vacJ* and wt cells (Figure 3). MS analysis of the OM fraction of pMK_*apfa*_s_-ACP_m_-*vacJ* cells assigned 49 total spectra (11 exclusive unique peptides, 19 exclusive unique spectra) to VacJ, as compared to 35 total spectra (10 exclusive unique peptides, 17 exclusive unique spectra) assigned to VacJ for the OM fraction of wt cells (Appendix A), indicating that the concentration of VacJ was about 29% higher for the pMK_*apfa*_s_-ACP_m_-*vacJ* OM fraction as compared to the wt. On the contrary, no significant differences in total spectra assigned to ApfA were detected between OM fractions of pMK_*apfa*_s_-ACP_m_-*vacJ* and wt strains (Appendix A). No spectra were assigned to the ACP_m_ tag in the OM fractions of both pMK_*apfa*_s_-ACP_m_-*vacJ* and wt strains (Appendix A).

## 3. Discussion

The in silico design of the ApfA_s_-ACP_m_-VacJ chimera was a multi-step process that can be divided in the identification of the three domains comprising the final protein: (i) an OM anchoring domain (ApfA_s_); (ii) a domain for post-expression localization and quantification of the chimeric antigen; (iii) a putative immunogen to be tested (VacJ). 

(i) ApfA is a fimbrial subunit protein involved in the initial stages of colonization of *A. pleuropneumoniae* whose role in pathogenesis and potential as an immunogen have previously been investigated [28,29,30,31,32,33]. As a well-characterized fimbrial protein, we hypothesized that ApfA would be exposed and anchored on *A. pleuropneumoniae* OM. This was subsequently supported by the in silico analysis performed in the current study, leading us to conclude that the ApfA_s_ domain would likely fold into an OM ‘’stem’’ (hence the ‘’S’’ in ApfA_s_) and could thus act as an anchoring domain within a chimeric open reading frame (ORF) designed to expose selected antigens on the surface of *A. pleuropneumoniae*. 

Due to the high conservation across Gram-negative bacteria of the general secretion pathway (GSP) domain identified in the ApfA sequence, we also hypothesized that a delivery system based on ApfA could offer a broader applicability and be employed for the selective OM enrichment of bacteria other than *A. pleuropneumoniae*. Additionally, the lack of significant homology between the ApfA_s_ domain and a selection of other proteins previously employed in membrane display strategies (OM proteins OmpA, LamB and PhoE; fimbrial and flagellar proteins FimH, PapA and flagellin; Autotransporter protein AidA) [34,35,36] supported the novelty of the strategy proposed. ApfA_s_ was thus selected as a suitable OM anchor.

(ii) The next feature of the chimeric antigen to be addressed was the need for post-expression localization and quantification. For this purpose, we relied on a method known as covalent labelling. This technique is based on the ability of certain enzymes to recognize specific peptide tags, catalyzing then the covalent transfer of substrates on the tagged protein [37]. The main reason behind the selection of this technique over autofluorescent-based detection here was the comparative advantage offered by covalent labelling when designing membrane-bound recombinant proteins. For covalent labelling in fact, both the enzyme and the substrate need to be externally supplied by the user. Contrary to autofluorescent-based detection, this ensures that only the exported fraction of the recombinant protein will be detected, as the labelling enzyme in particular will not be able to access intracellular compartments.

The selected tag for covalent labelling was the ACP-tag, a small tag recognized by the acyl carrier protein synthase (AcpS) for the covalent transfer of fluorescent derivatives of coenzyme A (CoA). [38]. Albeit relatively small, the full length of the ACP-tag is 77 amino acids, resulting in a 9-kDa domain. The sheer size of the ACP tag clearly hampers its usefulness for post-expression quantification and localization of recombinant immunogens, due to the likely alteration of the antigenic profile of the immunogen that the inclusion of such large domain would induce. Interestingly, it has been demonstrated that just eight residues of the ACP-tag are required for labelling by AcpS [39]. This shortened ACP-tag, defined here as ACP-mini (ACP_m_), presents two advantages over the full ACP-tag, both related to its smaller size. First and foremost, a domain of just eight amino acids is far less likely to affect trafficking and antigenicity of the engineered protein it is included in [20]. Secondly, the 24 nucleotides that encode for the ACP_m_ tag can be easily included as 5′ overhangs in the primers used during cloning, reducing the number of inserts needed to be stitched together for the creation of the recombinant ORF, thus increasing transformation efficiency. These considerations led us to select the ACP_m_ tag as a detection domain for the chimeric antigen discussed here.

(iii) The last step of the design consisted of identifying a native *A. pleuropneumoniae* immunogen to be tested. Ideally, this immunogen should possess the following characteristics: (1) proven immunogenicity, in order to provide immunological relevance to the chimera; (2) predicted OM subcellular localization, to increase the chances of seamless transport of the chimera through cellular membranes; (3) lack of toxicity, allowing high level of expression of the chimera in the host strain. 

Protein homologs belonging to the VacJ lipoprotein family are part of the conserved VacJ/Yrb ATP-binding cassette (ABC) transport system involved in phospholipid translocation through the OM [40]. We previously demonstrated that the VacJ-like homolog expressed in *A. pleuropneumoniae* strains presents a predicted OM subcellular localization, is easily expressible as recombinant protein and is highly immunogenic when administered in vivo as part of an immunization regime [32,33]. For these reasons, VacJ was selected as putative immunogen to be included in the chimera design described in this study.

Quantification by fluorescent labelling indicated that the ApfA_s_-ACP_m_-VacJ chimera is expressed and localized on the OM of pMK_*apfa*_s_-ACP_m_-*vacJ* cells. When using a fluorescent CoA-based labelling system though it is important to keep in mind that Acetyl-CoA is a major component of several metabolic pathways in bacteria, such as the biosynthesis of fatty acids and phospholipids. Accordingly, many Gram-negative bacteria actively import CoA to the cytoplasm and possess a native acetyl-transferase activity due to the expression of a homolog of the recombinant AcpS enzyme used for ACP-tag labelling [38]. This represents one of the main disadvantages when using ACP-tag labelling in bacterial strains, including *A. pleuropneumoniae*, as the accurate estimation of the expression level of the tagged recombinant protein can be skewed by a high background due to: (i) translocation to the cytoplasm of the fluorescent CoA substrate administered extracellularly during labelling; (ii) Incorporation of fluorescent CoA substrate into untagged proteins by native AcpS proteins. Both these processes represent a central part of the normal metabolism of unmodified bacterial strains and thus an inherent shortcoming of this labelling system. To overcome this problem, the ACP tag may need to be replaced with similar tags and fluorescent derivatives [41] in order to achieve a more stringent quantification of the levels of expression of the recombinant chimera.

To overcome this potential issue, protein assays were performed on the OM fractions of pMK_*apfa*_s_-ACP_m_-*vacJ* and wt cells to verify the actual concentration of the ACP_m_-VacJ chimera. Although SDS-PAGE showed no qualitatively detectable differences in banding patterns of the OM fractions of pMK_*apfa*_s_-ACP_m_-*vacJ* and wt cells, MS analysis suggested an increase of about 29% in the concentration of VacJ for the pMK_*apfa*_s_-ACP_m_-*vacJ* OM fraction as compared to the wt. This confirmed the level of enrichment of the ACP_m_-VacJ chimera previously observed by fluorescent labelling. Interestingly, no significant differences in ApfA concentration were detected between OM fractions of pMK_*apfa*_s_-ACP_m_-*vacJ* and wt strains, possibly due to the technical challenge posed by extraction and purification of trans-membrane protein domains such as the N-terminal of ApfA_s_ [42]. As expected, the small size of the ACP_m_ tag prevented its identification by the MS methodology employed in this study.

## 4. Materials and Methods 

### 4.1. In Silico Functional Prediction of ApfA Domains

A list of the genetic sequences, vectors and primers used in this study is provided in Table 1. The protein ApfA was selected for the identification of OM domains due to its well-characterized OM localization. The ApfA peptide sequence was analyzed by different online tools for functional prediction: (i) PANTHER™ scoring [43,44]; (ii) PHOBIUS [45]; (iii) I-TASSER [46]; (iv) PSORTb [7]. Nucleotides 1-205 from the *apfA* open reading frame were selected as ‘’stem’’ domain (*apfA*_s_) for OM anchoring of the chimeric antigen. The amino acid sequence of the *apfA*_s_ domain was analyzed by protein BLAST [47] to assess inter-species homology. Standard parameters were used unless otherwise stated.

### 4.2. Design of the apfA_s_-ACP_m_-vacJ Chimera

*apfA*_s_, ACP_m_ and *vacJ* protein sequences were combined into the chimeric *apfa*_s_-ACP_m_-*vacJ* ORF using the following rationale (N to C-terminal): ApfA_s_ as OM anchoring domain; ACP_m_ as reporter tag; VacJ as putative immunogen (Figure 4).

### 4.3. Construction and Transfer of the apfA_s_-ACP_m_-vacJ Expression Vector

*apfA*_s_ and *vacJ* nucleotide sequences (444 and 747 nucleotides respectively) were amplified from genomic DNA using AccuPrime Taq DNA high-fidelity Polymerase (ThermoFisher Scientific, Waltham, MA, USA). The pMK-express vector was linearized by reverse amplification using the same polymerase, in order to replace the native *GFPmut3* gene with the recombinant *apfa*_s_-ACP_m_-*vacJ* ORF by ligation-independent cloning (LIC). *apfA*_s_ and *vacJ* amplicons were joined with the linearized vector by three-point ligation-independent cloning, according to the In-Fusion protocol (Clontech Laboratories, Mountain View, CA, USA). The sequence of the ACP_m_ tag was introduced in frame between *apfA*_s_ and *vacJ* sequences by PCR-based tagging, including the 24 nucleotides of the tag into the 15 bp overhangs of *apfA*-Rev and *vacJ*-Fwd primers (Table 1). The resulting pMK_*apfa*_s_-ACP_m_-*vacJ* construct was transformed into *E. coli* Stellar competent cells (Clontech Laboratories, Mountain View, CA, USA) according to the In-Fusion protocol, with selection of recombinant clones on brain heart infusion (BHI) plates supplemented with 75 µg mL^−1^ kanamycin. Positive clones were analyzed by PCR and sequencing to verify the acquisition of the vector and the presence of the correct recombinant ORF. The pMK_*apfa*_s_-ACP_m_-*vacJ* construct was extracted and purified using a miniprep kit (Qiagen, Hilden, Germany) and subsequently transformed into *E. coli* electrocompetent S17-1 λpir cells, prepared and transformed as described in Reference [48]. Positive S17-1 λpir clones were then used for transferring the pMK_*apfa*_s_-ACP_m_-*vacJ* vector into *A. pleuropneumoniae* wt cells by conjugation, following the protocol described in Reference [49]. *A. pleuropneumoniae* pMK_*apfa*_s_-ACP_m_-*vacJ* clones were analyzed by PCR to verify the acquisition of the recombinant pMK_*apfa*_s_-ACP_m_-*vacJ* vector.

### 4.4. Outer Membrane Isolation

*A. pleuropneumoniae* serotype 8 MIDG2331 [50] wt and the pMK_*apfa*_s_-ACP_m_-*vacJ* transformed strain were inoculated into 5 mL of BHI broth containing 20 mg L^−1^ β-NAD (β nicotinamide adenine dinucleotide) (Merck Millipore, Burlington, Massachusetts, United States) in sterile 50 mL centrifuge tubes. Next, the cultures were incubated overnight (≥16 h) at 37 °C with agitation (200 rpm) in an aerobic atmosphere. 10 μL of the cultures were inoculated into 50 mL fresh BHI broth containing 20 mg L^−1^ β-NAD in 250 mL Erlenmeyer conical flasks. After incubation for 12–14 h, the cultures were adjusted to an OD_600_ of 1 and centrifuged for 1 h at 2600× *g*. The pelleted cells were then resuspended, washed once in 10 mmol L^−1^ HEPES of pH 7.4 and stored at −80 °C. After thawing, 15 µl of 1 mmol L^−1^ MgCl_2_ and standard units of DNAse I and lysozyme were added to the cells. The cells were lysed using a bead beater (Precellys Minilys, Bertin Technologies, Montigny-le-Bretonneux, France) (3 cycles of 3200× *g* for 3 min). The lysate was then centrifuged for 5 min at 29,000× *g* at 4 °C in a tabletop centrifuge to remove cell debris. The supernatant was moved to a fresh 2 mL tube and centrifuged for 1 h at 29,000× *g* at 4 °C to collect membranes. The pelleted membranes were resuspended thoroughly in 0.2 mL 10 mmol L^−1^ HEPES of pH 7.4. The inner membrane was then solubilized with 0.2 mL of 0.2% lauroyl sarcosine in 10 mmol L^−1^ HEPES (pH 7.4) for 30 min at room temperature. Following this, the solution was centrifuged for 1 h at 29,000× *g* at 4 °C to pellet the OM. The pellet was then washed once with 10 mmol L^−1^ HEPES (pH 7.4) and resuspended in 30 µl of 10 mmol L^−1^ HEPES.

### 4.5. SDS-PAGE

For SDS-PAGE, a standard protocol was applied. 10 µl of sample buffer, Laemmli, 2x concentrate loading buffer and 10 µl of supernatant and OM fractions of *A. pleuropneumoniae* serotype 8 MIDG2331 wt and pMK_*apfa*_s_-ACP_m_-*vacJ* strain were loaded onto 4–20% gradient SDS-Page gel (Bio-Rad). The Precision Plus Protein™ Dual Color Standard (Bio-Rad, Hercules, CA, USA) was included for molecular weight estimation. The proteins were run at 150 V for 45 min and then stained with Coomassie Blue. Imaging was performed using a Gel doc^™^ XR+ device with Image Lab^™^ software (Bio-Rad, Hercules, CA, USA).

### 4.6. Liquid Chromatography-Mass Spectrometry

Protein concentrations of inner and OM fractions were determined by Pierce™ bicinchoninic acid (BCA) Protein Assay Kit (ThermoFisher Scientific, Waltham, MA, USA). For liquid chromatography-mass spectrometry (LC/MS) analysis, all samples were adjusted to the protein concentration of 0.4 mg mL^−1^ in 10 mmol L^−1^ HEPES of pH 7.4. The samples were digested with 10 μg trypsin (sequencing grade) (Promega, Madison, WI, USA) overnight at 37 °C. The digestion was stopped by adding 5 μL 50% formic acid and the generated peptides were purified using a ZipTip C18 (Merck Millipore, Burlington, MA, USA) according to the manufacturer’s instructions and dried using a Speed Vac concentrator Plus (Eppendorf, Hamburg, Germany). The tryptic peptides were analyzed using an Ultimate 3000 nano-UHPLC system connected to a Q Exactive mass spectrometer (ThermoFisher Scientific, Waltham, MA, USA) equipped with a nano electrospray ion source.

### 4.7. Protein Identification by Scaffold Viewer

LC/MS data were analyzed using Scaffold^TM^ (Scaffold_4.8.9, Proteome Software Inc., Portland, Oregon, United States). The threshold for peptide identification was set at >95.0% probability (Peptide Prophet algorithm) [51] with Scaffold delta-mass correction. The threshold for protein identification was set at >99.9% probability in addition to at least one identified peptide. Protein probabilities were assigned by the Protein Prophet algorithm [52]. Proteins that contained similar peptides and which could not be differentiated based on tandem mass spectrometry (MS/MS) analysis alone were grouped to satisfy the principles of parsimony.

### 4.8. Flow Cytometry

Inoculation loops containing stored (–80 °C) *A. pleuropneumoniae* serotype 8 MIDG2331 [50] wt and the pMK_*apfa*_s_-ACP_m_-*vacJ* plasmid containing strain were inoculated into 5 mL of BHI broth containing 20 mg L^−1^ β-NAD in sterile 50 mL centrifuge tubes. Next, the cultures were incubated overnight (≥16 h) at 37 °C with agitation (200 rpm) in an aerobic atmosphere. 10 μL of the cultures were inoculated in 50 mL fresh BHI broth containing 20 mg L^−1^ β-NAD in 250 mL Erlenmeyer conical flasks. After incubation for 12–14 h, the OD_600_ of the cultures were adjusted to 0.1. Cells from 500 µl of the OD_600_ adjusted cultures were harvested (5000× *g*, 4 min) and re-suspended in 100 µl of a staining solution. For the preparation of the staining solution, 10^−2^ mol L^−1^ MgCl_2_, 0.625 nmol L^−1^ ACP synthase (NEB, Ipswich, MA, USA) and 0.25 nmol L^−1^ fluorescent CoA 488 (NEB, Ipswich, MA, USA) were added in BHI broth. Incubation was performed in a 1 mL Eppendorf tube in heating block at 37 °C for 1 h. The stained cells were harvested (5000× *g*, 4 min) and re-suspended in 200 µl of sterile filtered (0.22 µm sterile filters, Sartorius, Göttingen, Germany) Dulbecco’s phosphate-buffered saline (ThermoFisher Scientific, Waltham, MA, USA) and analyzed by flow cytometry (BD Accuri^®^ C6, BD Biosciences, San Jose, CA, USA). Following acquisition, the data obtained were analyzed using the FlowJo V10 software (BD Biosciences, San Jose, CA, USA). Three independent experiments were performed. For each data set, the statistical difference between the mean green fluorescence values of wt and pMK_apfas-ACPm-vacJ strains was determined by t-test (*p* < 0.01).

### 4.9. Fluorescence Microscopy

Cultures of *A. pleuropneumoniae* serotype 8 MIDG2331 wt and pMK_*apfa*_s_-ACP_m_-*vacJ* strain were prepared as described for flow cytometry. After staining, the cells were harvested (5000× *g*, 4 min) and re-suspended in 50 µl of filtered (0.22 µm sterile filters) Dulbecco’s phosphate-buffered saline (#141900, ThermoFisher Scientific, Waltham, MA, USA) and analyzed with fluorescence microscopy (Leica DM5000B, Leica Microsystems, Wetzlar, Germany). For the acquisition of images, phase contrast microscopy with 100× magnification and an oil-immersion objective lens were used. For fluorescence imaging, filter cube I3 with band pass 450-490 nm excitation filter, 510 nm dichromatic mirror and long pass 515 suppression filter were used. The cell cultures were mixed in a 1:1 ratio with 50% sterile glycerol on the glass slides just before the microscopy analysis in order to reduce the mobility of the cells for the acquisition of images. Three independent experiments were performed.

## 5. Conclusions

These results represent a valid proof of concept indicating that ApfA_s_ could be used as a fusion partner to direct translocation and exposure of small proteins on *A. pleuropneumoniae* OM. This would ideally result in the generation of recombinant strains which, once inoculated in the host as part of an immunization regime, could elicit a lasting humoral response against the antigens selected for ApfA_s_-mediated OM enrichment. Additionally, the cross-species conservation of some ApfA_s_ domains leads us to speculate that the expression platform described in this study could be used for the construction of recombinant chimeras aimed at antigen enrichment of the OM of other Gram-negative species of interest. Nonetheless, all these assumptions are preliminary in nature and further testing will be necessary to verify these hypotheses.

## Figures and Tables

**Figure 1 pathogens-09-01014-f001:**
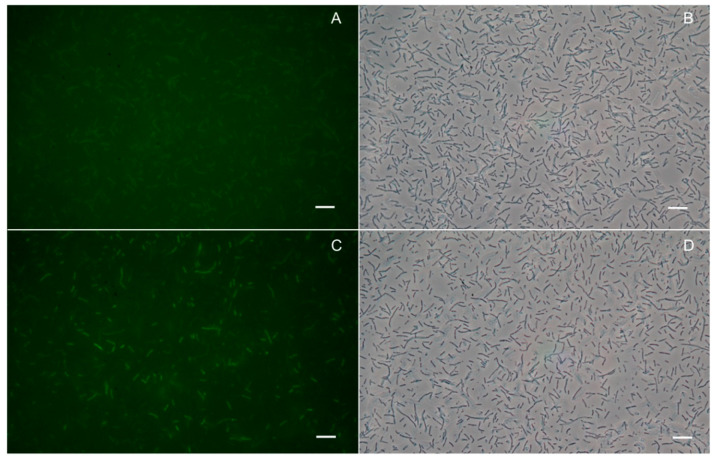
Fluorescence and non-fluorescence phase contrast microscopy images of *A. pleuropneumoniae* wild type (wt) (**A** and **B** respectively) and pMK_*apfa*_s_-ACP_m_-*vacJ* cells (**C** and **D** respectively) after staining with fluorescent CoA 488. Scale bars represent 10 μm. Results presented are representative of three independent experiments.

**Figure 2 pathogens-09-01014-f002:**
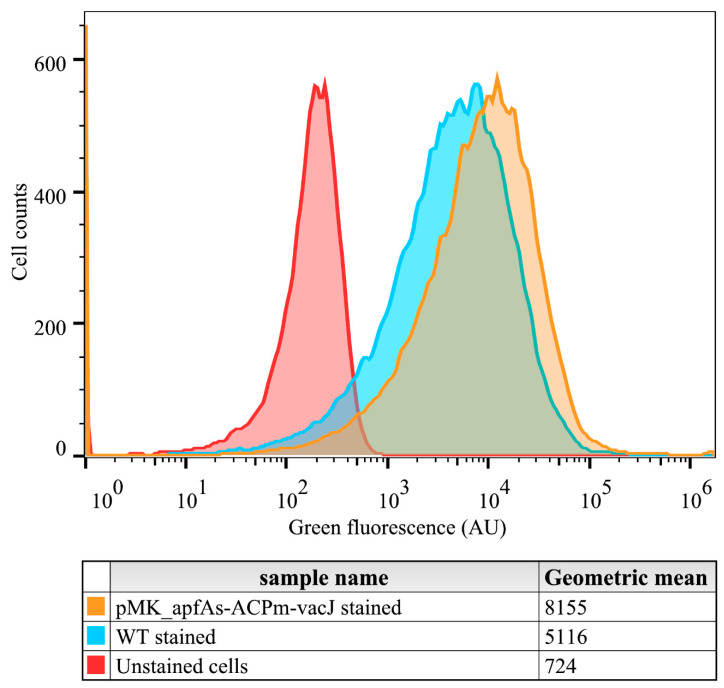
Fluorescence levels of *A. pleuropneumoniae* wt (blue) and pMK_*apfa*_s_-ACP_m_-*vacJ* cells (orange) after staining with fluorescent CoA 488 and analysis by flow cytometry. The histograms show the distribution of green fluorescence levels in the populations of each strain. The histogram of unstained cells of *A. pleuropneumoniae* wt is also presented (red). The geometric mean of the fluorescent cells of each sample is shown below the figure. Results presented are representative of three independent experiments. AU: arbitrary units. WT: wild type.

**Figure 3 pathogens-09-01014-f003:**
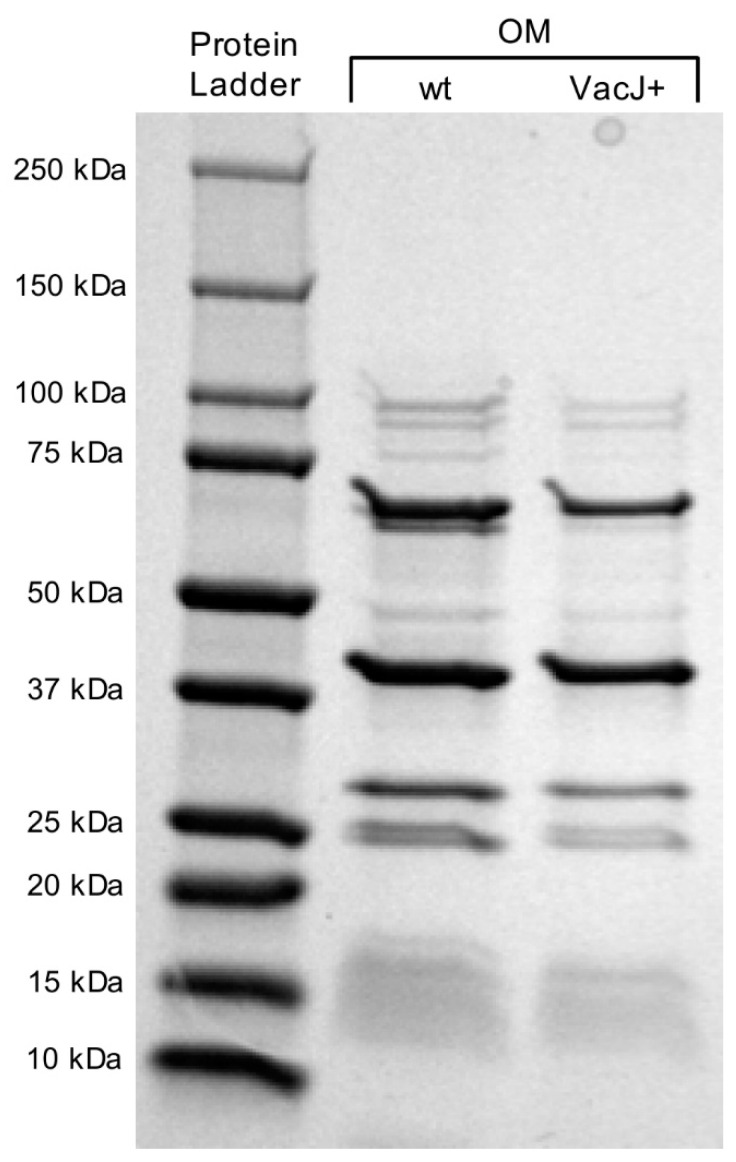
SDS-PAGE (sodium dodecyl sulfate polyacrylamide gel electrophoresis) analysis of outer membrane (OM) fractions of *A. pleuropneumoniae* serotype 8 MIDG2331 wt and pMK_*apfa*_s_-ACP_m_-*vacJ* cells. wt: wild type; VacJ+: pMK_*apfa*_s_-ACP_m_-*vacJ*; Protein ladder: Precision Plus Protein™ Dual Color Standards (Bio-Rad). The molecular weight of protein ladder bands is reported for reference.

**Figure 4 pathogens-09-01014-f004:**
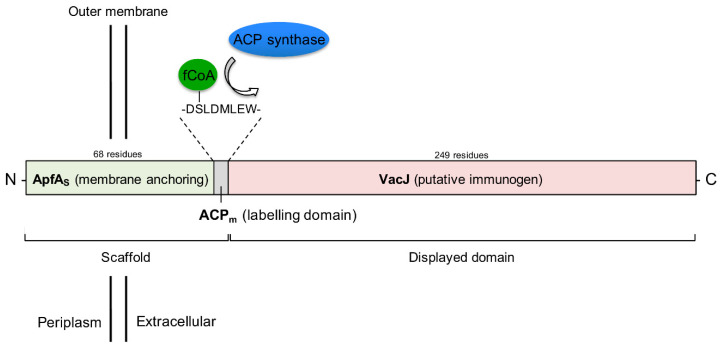
Functional outline of structure and predicted subcellular localization of the *apfa*_s_-ACP_m_-*vacJ* chimera. ACP_m_ amino acid sequence and ACP synthase target site on ACP_m_ are also shown. Relative domain size is represented to scale. ApfA_s_: ApfA stem domain; ACP_m_: ACP-mini tag; VacJ: putative immunogen; fCoA: fluorescent Acetyl-Coenzyme A. N-: N-terminal; C-: C-terminal.

**Table 1 pathogens-09-01014-t001:** Sequences and primers used in this study.

Denomination	Sequence (5′-> 3′)/Accession Number (GenBank)	Function
App 5b L20 *apfA* gene	CP000569.1	*apfA* ORF
App 5b L20 *apfA*_s_ locus	Nucleotides 1-205 from *A. pleuropneumoniae* 5b L20 *apfA*	*apfA* stem
App 5b L20 *vacJ* gene	CP000569.1	*vacJ* ORF
ACP_m_ tag	GATTCGCTTGATATGCTGGAGTGG	Indirect detection of expression
pMK-express vector	GQ334690.1	Naïve expression vector
pMK_*apfa*_s_-ACP_m_-*vacJ* recombinant vector	Full sequence provided in Appendix A	Recombinant expression vector
*apfA* Fwd	CATcAGTAAAGGAGAATGCAgAAgCTAAGTCTTATTCGA	*apfA* amplification
*apfA* Rev	CCAGCATATCAAGCGAATCTCCGGTGTTATATATGCAGATCTCG	
*vacJ* Fwd	GCTTGATATGCTGGAGTGGAAgTTAAAgCAATTAAGgTTAGTAGCC	*vacJ* amplification
*vacJ* Rev	CAATTCACTGGCCGTTCTGCTCCTTTGCCCTATCC	
pMK Rev	ACTTAGcTTcTGCATTCTCCTTTACTgATGGTCAATTCTC	pMK vector linearization
pMK Fwd	GGGCAAAGGAGCAGAACGGCCAGTGAATTGTAATACG	
pMK Fwd sequencing	CGCCAACCGATAAAACCTAC	Upstream sequencing
*vacJ* Rev sequencing	CTTTTTACCCTCGCCCTCT	
*vacJ* Fwd sequencing	GGCGTGATTATGTGCCGA	Downstream sequencing
pMK Rev sequencing	CAATACGCAAACCGCCTC	

Lower case letters indicate nucleotides modified from the target sequence in order to reduce the formation of primer secondary structures or satisfy the parameters for In-Fusion cloning.

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
