# Peer review of "Design and Characterization of a Novel Tool for the Antigenic Enrichment of Actinobacillus pleuropneumoniae Outer Membrane"

_pathogens, 2020, doi:10.3390/pathogens9121014_

Round 1

Reviewer 1 Report

The text of manuscript reflects quality study. Contents of all sections are appropriate and adequate. Materials and methods used in the study are adequately described. Results are generally well described and presented in the manuscript as well as discussion which is comprehensive. Conclusions were justified by the obtained results and correspond to the aim of the study. I made only some minor comments listed below.

Minor Comments:

  • Lines 19 and 22 - Abbreviations ApfAs and ACPm should be explained.
  • Line 110 - Is it possible to provide statistical verification of geometric mean difference to justify expression “significantly higher”?
  • Line 114 – “wt” should be explained herein (in order of appearance) not further in the Materials and Methods section. In my opinion in should be also explained in Fig. 1. and Fig. 2. as you did on Fig. 3.

Author Response

Reviewer 1: The text of manuscript reflects quality study. Contents of all sections are appropriate and adequate. Materials and methods used in the study are adequately described. Results are generally well described and presented in the manuscript as well as discussion which is comprehensive. Conclusions were justified by the obtained results and correspond to the aim of the study. I made only some minor comments listed below.

Minor Comments:

  • Lines 19 and 22 - Abbreviations ApfAs and ACPm should be explained.

Authors' Response: Done as requested.

  • Line 110 - Is it possible to provide statistical verification of geometric mean difference to justify expression “significantly higher”?

Authors' Response: A section detailing the statistical method used to analyze flow cytometry data has been added.

  • Line 114 – “wt” should be explained herein (in order of appearance) not further in the Materials and Methods section. In my opinion in should be also explained in Fig. 1. and Fig. 2. as you did on Fig. 3.

Authors' Response: Done as requested.

Reviewer 2 Report

An excellent paper on an original topic that could have significant applications in the field of vaccinology.

  • The Introduction explains very well the reasons and the issues of developing an alternative method for improving the screening of immunogen candidates.
  • The reasons why the ApfA OM of A. pleuropneumoniae has been chosen should be better explained in this section.
  • The Materials and Methods section is very much detailed.
  • The Results section is short but contains the relevant information.
  • The Discussion section addresses the major issues of the proposed technology.
  • The Conclusion is short. Maybe it should elaborate a bit about the kind of immune response the chimeric ApfAs-ACPm-VacJ antigen could elicit in the pig.

Author Response

Reviewer 2: An excellent paper on an original topic that could have significant applications in the field of vaccinology.

  • The Introduction explains very well the reasons and the issues of developing an alternative method for improving the screening of immunogen candidates.
  • The reasons why the ApfA OM of A. pleuropneumoniae has been chosen should be better explained in this section.

Authors' Response: Although this is a fair point, we believe that the explanation of the rationale behind the choice of using ApfA as putative anchor is better suited for the Discussion section. The Introduction section, as now formulated, serves more as a general overview on the potential and challenges of antigenic enrichment for vaccine development.

  • The Materials and Methods section is very much detailed.
  • The Results section is short but contains the relevant information.
  • The Discussion section addresses the major issues of the proposed technology.
  • The Conclusion is short. Maybe it should elaborate a bit about the kind of immune response the chimeric ApfAs-ACPm-VacJ antigen could elicit in the pig.

Authors' Response: A short paragraph has been added regarding the immunological potential of antigenic enrichment by the ApfAs delivery system proposed.